# Novel Opioids in the Setting of Acute Postoperative Pain: A Narrative Review

**DOI:** 10.3390/ph17010029

**Published:** 2023-12-25

**Authors:** Ashley Wang, Jasper Murphy, Lana Shteynman, Neil Daksla, Abhishek Gupta, Sergio Bergese

**Affiliations:** 1Department of Anesthesiology, Stony Brook University Hospital, Stony Brook, NY 11794, USA; ashley.wang@stonybrookmedicine.edu (A.W.); neilchristopher.daksla@stonybrookmedicine.edu (N.D.); abhishek.gupta@stonybrookmedicine.edu (A.G.); 2Renaissance School of Medicine, Stony Brook University, Stony Brook, NY 11794, USA; jasper.murphy@stonybrookmedicine.edu (J.M.); lana.shteynman@stonybrookmedicine.edu (L.S.); 3Department of Neurosurgery, Stony Brook University Hospital, Stony Brook, NY 11794, USA

**Keywords:** oliceridine, tapentadol, cebranopadol, dinalbuphine sebacate, nalbuphine, dual enkephalinase inhibitors, endomorphin-1 analog, acute postoperative pain

## Abstract

Although traditional opioids such as morphine and oxycodone are commonly used in the management of acute postoperative pain, novel opioids may play a role as alternatives that provide potent pain relief while minimizing adverse effects. In this review, we discuss the mechanisms of action, findings from preclinical studies and clinical trials, and potential advantages of several novel opioids. The more established include oliceridine (biased ligand activity to activate analgesia and downregulate opioid-related adverse events), tapentadol (mu-opioid agonist and norepinephrine reuptake inhibitor), and cebranopadol (mu-opioid agonist with nociceptin opioid peptide activity)—all of which have demonstrated success in the clinical setting when compared to traditional opioids. On the other hand, dinalbuphine sebacate (DNS; semi-synthetic mu partial antagonist and kappa agonist), dual enkephalinase inhibitors (STR-324, PL37, and PL265), and endomorphin-1 analog (CYT-1010) have shown good efficacy in preclinical studies with future plans for clinical trials. Rather than relying solely on mu-opioid receptor agonism to relieve pain and risk opioid-related adverse events (ORAEs), novel opioids make use of alternative mechanisms of action to treat pain while maintaining a safer side-effect profile, such as lower incidence of nausea, vomiting, sedation, and respiratory depression as well as reduced abuse potential.

## 1. Introduction

The management of postoperative pain is crucial in ensuring good outcomes for surgical patients. However, results from national surveys in the United States reveal that over 80% of patients undergoing surgery complain of inadequately treated postoperative pain [1,2]. Poorly controlled acute postoperative pain has been linked to a myriad of complications, from interfering with patients’ ability to participate in rehabilitation activities and prolonging length of hospital stays to increasing postoperative morbidity and mortality [1]. Although opioids have historically been the mainstay for treating postoperative pain, they have limitations due to opioid-related adverse events (ORAEs) from nausea, vomiting, and pruritus to sedation and respiratory depression [3]. In addition, long-term use of opioids may lead to dependence and addiction. In this review, we present several novel opioids with unique mechanisms of action that deliver potent analgesia with an improved safety profile.

### 1.1. Endogenous Opioid Receptors and Ligands

A strong understanding of the pharmacology of opioids is essential for healthcare professionals to optimize their therapeutic benefits while minimizing the associated risks. Opioids exert their effects by interacting with opioid receptors, which belong to a group of G-protein-coupled receptors (GPCRs) responsible for facilitating pain relief in both the central and peripheral nervous systems [4]. GPCRs have seven transmembrane domains and are an evolutionarily conserved group of receptors that can interact with a wide array of signaling molecules to perform many functions in the human body [5,6].

The endogenous opioid system consists of three main receptor types: mu (μ), delta (δ), and kappa (κ) receptors. These receptors are widely distributed throughout the central nervous system (CNS) and peripheral tissues. They are primarily responsible for mediating the analgesic, sedative, euphoric, and potentially addictive effects of opioids [7].

#### 1.1.1. Mu (μ) Receptor

Mu-opioid receptors (MORs) are the primary target of opioid analgesics. Three subtypes of mu receptors have been identified: μ1, μ2, and μ3. Activation of mu receptors leads to an intricate cascade of signaling events that ultimately results in pain suppression, euphoria, respiratory depression, and other physiological and behavioral effects associated with opioids [7].

Pain Suppression: Upon activation, presynaptic MORs inhibit the release of excitatory neurotransmitters involved in pain signaling, such as substance P and glutamate. Postsynaptic receptors hyperpolarize postsynaptic neurons, leading to inhibition of afferent transmission. This modulation of pain pathways reduces the transmission of pain signals and alters pain perception, resulting in pain suppression [8].

Euphoria: Activation of MORs stimulates the release of dopamine, a neurotransmitter associated with pleasure and reward through its inhibition of GABA secretion in the reward pathway of the brain. This release of dopamine contributes to the euphoric and addictive effects experienced by individuals taking opioids [7].

Respiratory Depression: Respiratory depression is a potentially dangerous side effect of opioid use. MOR stimulation in respiratory centers can affect the action of the cerebral cortex, thalamus, peripheral chemo and baroreceptors, and mechanoreceptors. Depression of pathways driving respiration results in irregular, slow breathing with the potential progression to hypoxia and hypercapnia [7].

Inhibitory Neuronal Signaling: Peripheral opioid receptor activation, primarily of MORs, opens G-protein-coupled inwardly rectifying K^+^ (GIRK) channels. This leads to hyperpolarization of neuronal cell membranes and inhibition of action potential propagation [9]. One study in mice found that this K^+^ channel is both necessary and sufficient for peripheral opioid-mediated analgesia [9].

#### 1.1.2. Mu Receptor Endogenous Ligands

The main endogenous ligands for mu receptors are the enkephalins, β-endorphin, and dynorphins. Enkephalins which consist of met- and leu-enkephalin are widely distributed in the CNS and have many physiological effects. However, their main function is thought to relate to analgesia, stress response regulation, and peristalsis modulation [10]. β-endorphin is a peptide produced by the arcuate nucleus of the hypothalamus and secreted through the pituitary gland. It is produced during stress and exercise, acting to reduce muscular fatigue, stimulate glucose uptake, induce euphoria, and inhibit post-exercise pain [8]. Dynorphins, encoded by the prodynorphin gene, are widely distributed in the CNS and are involved in learning and memory, emotional control, stress response, and pain [11].

#### 1.1.3. Delta (δ) Receptor and Ligands

Delta receptors are primarily involved in pain modulation and are distributed throughout the CNS. They are expressed in primary afferents, the very first step of pain processing pathways, and can activate GIRK channels along with mu receptors [12]. Activation of delta receptors can result in analgesia, although their contribution to the overall effects of opioids is less prominent than that of MORs.

The main endogenous ligand for delta receptors is the enkephalin peptide, which binds to both delta and mu receptors. The affinity of enkephalins for delta receptors is higher than that for mu receptors [7].

#### 1.1.4. Kappa (κ) Receptor and Ligands

Kappa receptors are widely distributed in the CNS. Activation of kappa receptors can produce analgesia, dysphoria, and sedation as well as modulate other physiological functions such as stress, mood, inflammation, and remyelination. Kappa receptor activation is also associated with decreased reward and dependence liability, making kappa agonists an area of interest for potential therapeutic applications. However, clinical use of KOR agonists is limited by adverse effects such as dysphoria, aversion, and sedation [13].

Dynorphins are the primary endogenous ligands for kappa receptors. Activation of kappa receptors by dynorphins can lead to dysphoric effects which may contribute to the lower abuse potential associated with kappa agonists [13].

#### 1.1.5. Nociceptin Opioid Peptide (NOP) Receptor

The nociceptin opioid peptide (NOP) receptor, also known as the orphanin FQ receptor, is another type of opioid receptor that differs from the mu, delta, and kappa receptors due to its binding specificity and functional effects. It is also a GPCR and shares similar homology to other opioid receptors. The NOP receptor specifically interacts with the endogenous peptide nociceptin or orphanin FQ, rather than the traditional opioid peptides like endorphins, enkephalins, and dynorphins [14,15].

Activation of the NOP receptor by nociceptin can have complex effects on the nervous system, including modulation of pain perception, regulation of stress responses, and involvement in mood regulation [14]. While it is considered an opioid receptor due to its structural similarities and interactions with some opioid ligands, its distinct characteristics set it apart from the traditional mu, delta, and kappa receptors.

A number of factors have the potential to influence opioid signaling and require further study. For example, positron emission tomography scans have shown that MOR availability in older individuals was increased in frontotemporal regions but decreased in the amygdala, thalamus, and nucleus accumbens. In addition, reproductive-age females were found to have higher MOR availability than males in multiple regions, but the effects of age were stronger in males [16]. Studies have also demonstrated that the analgesic effects of mu, delta, and kappa receptor agonists are more pronounced in inflamed tissues compared to non-inflamed tissues [17]. Inflammation can lead to alterations in opioid receptor expression at the mRNA and protein levels, with mu receptors consistently showing upregulation in response to inflammation. Additionally, different kinases have the ability to phosphorylate intracellular regions of opioid receptors, thereby enhancing the binding of arrestin molecules. The resulting formation of arrestin–opioid receptor complexes induces receptor desensitization by impeding G protein-coupling while also enabling receptor internalization through clathrin-dependent pathways [18]. Opioid receptors are also subject to epigenetic regulation and alteration at mRNA and protein levels [19].

### 1.2. Conventional Opioids and Mechanism of Action

Naturally derived products offer the possibility of developing novel compounds [20]. The history of conventional opioids dates back to the use of opium, a substance derived from the opium poppy plant *Papaver somniferum*. Opium contains various alkaloids, including morphine and codeine, which have potent analgesic properties. After isolating morphine from opium in the nineteenth century, chemists began modifying its chemical structure to create semi-synthetic and fully synthetic opioids. By introducing chemical modifications, they aimed to enhance specific properties such as potency, duration of action, or reduced side effects [10].

Common conventional opioids include morphine, codeine, oxycodone, hydrocodone, and fentanyl. Each of these opioids interacts with opioid receptors, predominantly MORs, to produce analgesia and other effects. It is important to note that opioids can also bind to other opioid receptor subtypes (delta, kappa, and NOP), contributing to their overall effects and potential side effects [10].

### 1.3. Opioid-Related Adverse Effects and the Opioid Crisis

Prescription of opioids for acute or chronic pain is common and even customary. However, the usage of opioids for long durations becomes contentious. In addition to the risk of addiction, opioid usage is known to cause a number of adverse effects.

Frequent side effects of opioid administration include sedation, dizziness, nausea, vomiting, constipation, physical dependence, tolerance, and respiratory depression. Physical dependence and addiction may hinder proper prescription and lead to insufficient pain management. There are also less common side effects, including delayed gastric emptying, hyperalgesia, immunologic and hormonal dysfunction, muscle rigidity, and myoclonus. Among these side effects, constipation and nausea are the most prevalent. Constipation, in particular, has a notably high occurrence rate that is challenging to manage. Both constipation and nausea can be severe enough to warrant stopping opioid usage, contributing to inadequate pain relief and suboptimal dosing [21].

The side effect of respiratory depression is of greatest concern because of the increased risk of fatality. In fact, a study of eight individuals found that mild to moderate respiratory depression occurs at morphine concentrations that do not cause any analgesic effect [22]. The PRODIGY trial looked at continuous capnography and oximetry of 1335 patients receiving parenteral opioids and found that 46% of patients experienced one or more episodes of respiratory depression. These patients had a mean length of hospital stay three days longer compared to that of those without episodes of respiratory depression [23]. The Joint Commission has designated respiratory depression as the most serious adverse event that can occur with the use of opioid medications for analgesia [24].

A 2018 retrospective study of 135,379 patients receiving opioids after surgical and endoscopic procedures showed that over 10% experienced at least one ORAE. In addition, ORAEs were associated with a 2.9% increase in absolute mortality, a 1.6-day increase in length of stay, and a USD 8225 increase in cost [25]. Although conventional opioids such as morphine may be cheaper, novel opioids may be able to offset their cost with a reduced incidence of ORAEs. A more targeted approach can further increase cost-effectiveness by using them only in high-risk patients, such as those with advanced age, obesity, or sleep apnea.

Opioid use disorder involves the misuse of prescribed medications or substances obtained illicitly. The disorder is considered a chronic illness often characterized by relapse. While treatments are available to halt usage or prevent relapse (including methadone), opioid use disorder is associated with significant rates of morbidity and mortality. Because of its high costs, both societally and fiscally, opioid misuse is deemed a public health crisis.

The use of opioids in the United States has been described as an epidemic. Drug overdose deaths have nearly tripled in the time between 1999 and 2014. In fact, among the 47,055 drug overdose deaths of 2014, 60.9% involved the use of an opioid [26]. In the 2015 National Survey of Drug Use and Health (NSDUH), it was estimated that 91.8 million U.S. civilian adults had used prescription opioids, 11.5 million had misused them, and 1.9 million had an opioid use disorder [27]. In the 2019 NSDUH, 5.7 million Americans were estimated to have used heroin at some point in their lives [28]. Another study determined that heroin use nearly doubled, increasing from 0.17% to 0.32% between the years 2002 and 2018 [29].

According to the information gathered from the Joint Commission’s Sentinel Event database, about 11% of adverse events linked to opioids in hospital settings between 2004 and 2011 are associated with opioid overdose, drug–drug interactions, or adverse drug reactions. However, it is possible that the true incidence could be greater than reported. Although the opioid crisis is well documented and benefits from increased awareness, opioid overdoses continue to claim the lives of a significant number of Americans [24].

## 2. Novel Opioids

Here, we present several novel opioids with the potential to deliver powerful analgesia while maintaining an optimal safety profile through unique mechanisms of action. A summary of each novel opioid can be found in Table 1.

### 2.1. Oliceridine

Oliceridine is a novel MOR agonist with biased ligand activity that preferentially stimulates G protein-coupling while downregulating the recruitment of β-arrestin, allowing it to limit ORAEs [44]. Oliceridine (brand name Olinvyk) was first approved by the FDA in 2020 to treat acute pain severe enough to warrant intravenous (IV) opioids when alternatives are inadequate [45]. It can be delivered in intermittent boluses or via patient-controlled analgesia (PCA). The FDA recommends administering a starting dose of 1.5 mg, with subsequent PCA demand doses of 0.35 mg to 0.5 mg with a 6 min lockout interval; the maximum total daily dose should not exceed 27 mg [45].

Oliceridine, an amine compound, has an onset of analgesia within 1 to 2 min, a peak effect at 6 to 12 min, and a half-life of 1.3 to 3 h [45]. EC_50_ was estimated as 10.1 ng/mL [46]. The chemical structure is shown in Figure 1. It is primarily metabolized in the liver by CYP3A4 and CYP2D6 P450 enzymes into inactive metabolites that are subsequently eliminated renally (70%) or fecally [45]. Comparing healthy subjects to those with end-stage renal disease (ESRD) and those with mild or moderate hepatic impairment, Nafziger et al. found no significant difference in oliceridine clearance. As such, no dosing adjustments are warranted for these patients [47]. However, they found that patients with severe hepatic impairment may benefit from reduced initial dosing and less frequent subsequent doses due to delayed clearance of the drug [45,47].

Oliceridine leads to adverse reactions similar to those seen in traditional opioids, including nausea, vomiting, dizziness, headache, constipation, pruritus, hypoxia, and respiratory depression [45]. In addition, two studies found some QTc prolongation through an unknown underlying mechanism [45].

#### 2.1.1. Clinical Studies

In a phase I randomized, double-blind crossover study, Soergel et al. compared the safety, tolerability, and analgesia (as determined by the cold pain test) of oliceridine (1.5 mg, 3 mg, and 4.5 mg), morphine (10 mg), or placebo in 30 healthy subjects. The results showed that all three doses of oliceridine produced analgesic effects, with 3 mg and 4.5 mg having higher peak analgesia and faster onset compared to morphine. Reduction in respiratory drive was present but transient in oliceridine groups at all doses, whereas that in morphine was more persistent. In addition, severe nausea was common after morphine and oliceridine at 4.5 mg but less frequent at a 1.5 mg or 3 mg dose. Overall, the findings determined 3 mg oliceridine to have the best analgesic efficacy while minimizing the incidence of nausea and vomiting [30].

Viscusi et al. conducted a phase II randomized controlled trial (RCT) of 144 patients with moderate or severe acute postoperative pain after bunionectomy. Patients were given oliceridine (1, 2, or 3 mg every 3 h), morphine (4 mg every 4 h), or placebo. The results showed that 2 and 3 mg oliceridine both led to significantly lower pain intensity as assessed by a numeric rating scale (NRS) than placebo over 48 h, with meaningful pain relief within 5 min and significantly greater pain relief after the first dose compared to morphine. Furthermore, no serious adverse events were reported with oliceridine use aside from dose-related ORAEs such as nausea, vomiting, dizziness, and headaches—similar to those found with morphine use [31]. These findings demonstrating the rapid analgesic efficacy and tolerability of oliceridine were reflected in another phase II double-blind RCT by Singla et al. of 200 patients with moderate to severe pain after abdominoplasty. Unlike the previous study, oliceridine was administered via PCA. They found significant reduction in pain score over 24 h for oliceridine compared to placebo, providing analgesia at a similar level as morphine. The oliceridine regimen also had a faster onset and meaningful pain relief within 0.3 h compared to 1 h for the morphine regimen. Finally, there were significantly fewer nausea, vomiting, and respiratory events (e.g., hypoxia, bradypnea, hypoventilation) reported with oliceridine compared to morphine [48]. Both phase II studies were split into two parts with an interim analysis in between to help determine the dosing regimen for the second part [31,48].

The phase III multicenter APOLLO RCTs continued to investigate the effectiveness of oliceridine PCA compared to morphine in patients who underwent bunionectomy (APOLLO-1, 389 patients) and abdominoplasty (APOLLO-2, 401 patients) [32,33]. Both studies demonstrated higher pain relief for oliceridine compared to placebo. APOLLO-1 noted fewer gastrointestinal adverse events with lower oliceridine demand doses (0.1 and 0.35 mg), whereas the higher demand dose of 0.5 mg had a similar level of side effects as the morphine regimen. Reports of lethargy and syncope were reported in the 0.5 mg oliceridine demand dose in APOLLO-2; however, the symptoms resolved without further complications.

Finally, ATHENA was a phase III multicenter, open-label study that aimed to evaluate how oliceridine may be used in real-world conditions. Unlike the previous studies, Bergese et al. expanded enrollment for the trial to not only postoperative patients but also non-surgical patients with painful medical conditions. In addition, they also included populations at high risk for ORAEs who were excluded from previous studies, such as those with advanced age, obesity, and sleep apnea. ATHENA also allowed multimodal analgesia, with oliceridine being delivered by bolus or PCA. With over 760 patients, ATHENA showed findings consistent with many prior studies: oliceridine is fast-acting and effective in reducing NRS pain score within 30 min of delivery. The most common adverse events were nausea (31%), constipation (11%), and vomiting (10%)—most of which were mild (37%) or moderate (25%) in severity. Only 33% of adverse events were considered possibly or probably related to oliceridine. Overall, oliceridine was deemed safe and well tolerated for pain management [34].

#### 2.1.2. Potential Advantages

Since oliceridine can selectively activate G-protein signaling associated with analgesia while simultaneously downregulating β-arrestin recruitment linked to ORAEs, it has potential for a widened therapeutic window. An exploratory analysis of the APOLLO studies by Beard et al. showed that when adjusted for analgesic effect, the odds of achieving “complete GI response” (i.e., no vomiting or use of rescue antiemetic) was two to three times higher with oliceridine than with morphine [49].

On the other hand, the APOLLO trials could not conclusively determine whether oliceridine has a lower incidence of opioid-induced respiratory depression (OIRD). Subsequently, Ayad et al. conducted an analysis of the APOLLO studies to specifically examine dosing interruption due to respiratory events and its cumulative duration as surrogate markers of OIRD. They found lower and shorter duration of dosing interruption with oliceridine compared to morphine [50].

Further analysis of ATHENA, which did not exclude patients at high risk of respiratory depression, also strived to investigate OIRD with oliceridine use. A retrospective chart analysis by Bergese et al. showed significantly lower incidence of OIRD events for patients receiving oliceridine compared to those on conventional opioids (8.0% vs. 30.7%) [51]. Future studies are warranted to better understand the association between oliceridine use and OIRD, including the ongoing phase IV VOLITION study to be completed in 2025 [44].

### 2.2. Tapentadol

Tapentadol (brand name Nucynta) is a centrally acting opioid analgesic that demonstrates both MOR agonist activity as well as norepinephrine (NE) reuptake inhibition [52,53,54]. Its chemical structure is shown in Figure 2. Tapentadol was first approved by the FDA in 2008 as an immediate-release (IR) oral tablet to treat opioid-requiring moderate to severe pain when alternatives are inadequate. Its extended-release (ER) formulation was later approved in 2011, with its use expanded for the management of neuropathic pain associated with diabetic peripheral neuropathy [52,53,54]. The FDA recommends starting tapentadol at a low dosage to match the treatment goals of individual patients before titrating to higher doses. The maximum total daily dose is 600 mg and 500 mg for IR and ER, respectively.

Tapentadol has an onset of analgesic effect of 30 min and a half-life of 4 h. It reaches peak serum concentration within an hour after administration [52]. EC_50_ was similar for both targets (1.8 μM for MOR, 2.3 μM for noradrenaline transporter) [55]. A total of 97% of the drug is metabolized by *O*-glucuronidation in the liver into inactive metabolites and subsequently excreted via the kidneys [52]. This is in contrast to tramadol, which requires metabolism by CYP2D6 into a potent metabolite in order to produce analgesia [35].

Adverse reactions commonly associated with tapentadol include nausea, dizziness, vomiting, and somnolence in adults (incidence ≥ 10%) or vomiting, constipation, nausea, pruritus, and pyrexia in pediatric patients age 6 or over (incidence > 5%). Patients taking tapentadol must be monitored closely for life-threatening respiratory depression, serotonin syndrome, adrenal insufficiency, severe hypotension, and sedation. Those at risk for adverse outcomes with tapentadol use include patients who are elderly, cachectic, or debilitated as well as those with chronic pulmonary disease or impaired consciousness. Tapentadol is contraindicated in patients with impaired pulmonary function (e.g., significant respiratory depression, acute or severe bronchial asthma), paralytic ileus, and concomitant use of monoamine oxidase inhibitors (MAOIs) or use within 14 days [52].

#### 2.2.1. Mechanisms of Action and Preclinical Studies

A 2007 study by Tzschentke et al. examined tapentadol binding to various receptors in both rat and human cells. The results showed that tapentadol has a K_i_ value (indicator of binding affinity) of 0.096 μM for MOR, which is an approximately 10-fold stronger affinity compared to delta- and kappa-opioid receptors (0.97 μM and 0.91 μM, respectively) and a nearly 50 times lower affinity compared to morphine for MOR (0.0022 μM). Tapentadol binds human recombinant MOR at a similar level as rat receptors at 0.16 μM. In human transporter binding assays, the results showed tapentadol to have K_i_ values of 8.80 μM and 5.28 μM for NE and serotonin (5-HT) transporters, respectively, whereas no binding was determined in morphine. Despite higher affinity of tapentadol for 5-HT over NE transporters, functional assays revealed more specific and selective synaptosomal uptake inhibition of NE (0.48 μM) compared to 5-HT (2.37 μM). This suggests that stronger binding of ligands to receptors does not automatically translate to functional significance. In other words, receptor binding does not always equate to efficacy [56].

In addition to assessing binding affinity to receptors and functional uptake inhibition, Tzschentke et al. also studied several rat and mouse pain behavioral models to evaluate the analgesic efficacy of tapentadol, including the hot plate test, tail-flick test, mustard oil-induced visceral pain, spinal nerve ligation (SNL), and writhing response. The authors found that despite having a 50-fold lower binding affinity to MOR, tapentadol delivered analgesia at a level only 2 to 3 times lower overall compared to morphine. This hints at the key role of NE reuptake inhibition in tapentadol to relieve pain, as demonstrated in the SNL model in which the analgesic effect of tapentadol was only moderately attenuated by naloxone (a MOR antagonist) whereas the analgesic effect of morphine was completely blocked by the antagonist. Finally, using the chronic constriction injury (CCI) model of neuropathic pain, Tzschentke et al. found a delayed tolerance development to the analgesic effect of tapentadol (18 days to onset of tolerance and 51 days to complete tolerance) compared to morphine (immediate onset of tolerance and 21 days to complete tolerance) [56].

#### 2.2.2. Clinical Studies

Tapentadol has been compared to other opioids for the management of acute pain in perioperative settings. In a systematic review of 13 studies and 1 abstract (including a total of 12,814 patients), Wang et al. examined the safety and efficacy of tapentadol IR after a variety of surgeries [35]; these included bunionectomy, cardiac surgery, dental procedures, total hip replacement, and abdominal hysterectomy. Results from the quantitative meta-analysis found that the lowest dose of tapentadol IR (i.e., 50 mg) was associated with less pain control, whereas higher doses (i.e., 75 mg or 100 mg) delivered a similar level of analgesia as oxycodone IR [35,57].

Qualitative synthesis by Wang et al. further examined the effect of tapentadol compared to other opioids. A phase II single-dose, double-blind RCT in 400 patients undergoing dental surgery found that higher dosage of tapentadol IR (200 mg), but not low dosage (100 mg), provided greater analgesia than morphine sulfate (60 mg) [58], whereas phase III multicenter, double-blind RCTs for abdominal hysterectomy (832 patients) and bunionectomy (285 patients) [57] showed similar levels of analgesia for tapentadol and morphine. On the other hand, comparing tapentadol and tramadol in 60 patients, Iyer et al. showed better pain control with tapentadol after cardiac surgery [59], while Moorthy et al. found no difference in pain score in an RCT of 100 patients experiencing acute osteoarthritic knee pain after one week of treatment [60]. Although it might not offer significantly improved analgesia compared with other types of opioids (e.g., oxycodone, morphine, tramadol), tapentadol appears to be better tolerated with lower incidence of gastrointestinal adverse events. Wang et al. acknowledged potential biases of the studies included in the analysis and noted that despite the variety of surgeries, the overrepresentation of minor procedures could pose a challenge in translating the results to major surgeries [35].

#### 2.2.3. Potential Advantages

Due to its unique dual mechanism of action, tapentadol is a powerful analgesic that makes use of the synergistic interaction of two targeted receptors to address both nociceptive pain (i.e., inflammatory process from tissue damage) via MOR agonism and neuropathic pain via NE reuptake inhibition—an advantage over traditional MOR-activating opioids. Fortunately, the combined effects of the two mechanisms do not translate to a higher burden of adverse effects [61]. Instead, this synergy allows for less mu-opioid activity, leading to an improvement in ORAEs. This is supported by a systematic review by Freynhagen et al. in patients with moderate or severe chronic pain that found that tapentadol caused less nausea and constipation than other opioids and less dizziness and somnolence than oxycodone and oxymorphone [62]. Furthermore, an exploratory study by van der Schrier et al. compared the effects of tapentadol and oxycodone on the ventilatory response to hypercapnia in healthy volunteers. While tapentadol still produced respiratory depression, the 100 mg dose may have some advantage over 20 mg oxycodone, warranting further study [63].

Tramadol, similar to tapentadol, is an opioid analgesic with a mixed mechanism of action of weak MOR agonism as well as 5-HT and NE reuptake inhibition [54,64]. As a result, it is also considered safer than conventional opioids due to its improved adverse effect profile. However, whereas tramadol relies on metabolism by the polymorphic CYP2D6 enzyme into an active metabolite to achieve analgesia, tapentadol as a parent drug is in itself an active compound. Due to its lower serotonergic effect, tapentadol theoretically has a lower risk of causing serotonin syndrome compared to tramadol [64]. However, there have been reports of serotonin syndrome when tapentadol is used in conjunction with serotonergic antidepressants [65], though current research in the literature on the phenomenon is limited due to inadequate characterization of adverse events or lack of distinction of patients who take serotonergic medications from those who do not [66].

Because of its MOR agonist activity, the abuse risk of tapentadol must be carefully considered. Although the extended-release formulation of tapentadol is difficult to crush and dissolve and, thus, less likely to be misused, there have been reports of immediate-release tablets being used recreationally after being crushed and injected, at times leading to death [54,67]. Results from a study that utilized data from the National Addictions Vigilance Intervention and Prevention Program found the abuse potential of tapentadol to be higher than that of tramadol and similar to that of hydrocodone but lower than that of other strong opioids (e.g., morphine) [54,68]. Overall, tapentadol can be a safe alternative to typical opioids, especially in hospital settings and for short durations [64].

### 2.3. Cebranopadol

Cebranopadol is a first-in-class potent opioid receptor agonist predominantly acting at the MORs and NOP receptors with lower activity at delta and kappa receptors (MOR: K_i_ = 0.7 nM and EC_50_ = 1.2 nM; NOP: K_i_ = 0.9 nM and EC_50_ = 13 nM; delta: K_i_ = 18 nM and EC_50_ = 110 nM; kappa: K_i_ = 2.6 nM and EC_50_ = 17 nM) [69,70]. It is a spiroindole derivative of the benzenoid class and displays promising experimental results in a range of animal models and clinical trials [15]; the chemical structure is shown in Figure 3. Cebranopadol displays a synergistic (rather than additive) effect of the activation of NOP and classical opioid receptors [71].

Kleideiter et al. examined the pharmacokinetic characteristics of cebranopadol through noncompartmental methods in six phase I clinical trials in healthy subjects. They found that IR cebranopadol reaches maximum concentration after 4 to 6 h and has a half-value duration of 14–15 h. With once-daily dosing, the time to steady state was approximately 2 weeks, and the peak–trough fluctuation was low. Kleideiter et al. concluded that these pharmacokinetic parameters make cebranopadol a promising therapeutic option for chronic pain [72].

#### 2.3.1. Preclinical Studies

In preclinical studies, cebranopadol has been shown to provide appropriate analgesia through peripheral and central administration in nociceptive and neuropathic pain [70], anti-hypersensitive effects in a rat model of arthritic pain [73], blockage of visceral pain [74], and blockage of pain in the trigeminal nerve distribution [75]. Additionally, Schiene et al. found that dual activation of NOP and mu receptors contributes to the anti-hypersensitive effect of cebranopadol in rat models [73].

There are also promising results regarding its reduced side effects and decreased use dependence. Researchers found that cebranopadol exhibits reduced respiratory depression in both adult rhesus monkeys and rats compared to fentanyl [76,77]. Through use of selective antagonists, Linz et al. mechanistically demonstrated that attenuation of respiratory depression in cebranopadol can be attributed to NOP receptor agonism which counteracts side effects resulting from MOR agonism [77]. Other studies indicate that cebranopadol has reduced physical dependence compared to other opioids. Tzschentke et al. conducted a direct comparison study, demonstrating that cebranopadol leads to a reduced occurrence of physical withdrawal symptoms compared to morphine [78]. This finding was observed across various dosages and administration periods in both mice and rats. Ruzza et al. used an elegant knockout experiment to compare symptoms of opioid withdrawal in mice. No disparity was observed in mice treated with morphine. However, cebranopadol administration induced more pronounced withdrawal symptoms in knockout mice. This finding suggests that NOP activation plays a key role in mitigating the physical dependence of cebranopadol [79].

Multiple studies have demonstrated that cebranopadol reduces cocaine self-administration and addiction-like behaviors in rats without directly influencing the pharmacokinetics of cocaine [80,81,82,83]. Given these results and its tolerability in humans, cebranopadol may have potential as a novel therapy for cocaine addiction. These preclinical findings highlight the potential of cebranopadol to offer reduced side effects.

#### 2.3.2. Clinical Studies and Potential Advantages

Clinical studies have reaffirmed many results from animal research. In a phase I study, Dahan et al. examined the respiratory effect of 600 mcg cebranopadol on 12 healthy volunteers. Although cebranopadol produced many of the effects expected of an opioid, respiratory depression was less severe and had a ceiling due to activation at the NOP receptor. This suggests an advantage over conventional opioids, which may produce apnea at high concentrations [36]. In addition, another phase I study in 48 healthy nondependent recreational opioid users found a reduced peak effect of drug-liking in cebranopadol relative to hydromorphone, suggesting reduced recreational potential for the drug [37]. Finally, Scholz et al. conducted a phase IIa multicenter, double-blind RCT in 258 patients undergoing bunionectomy. They found that the administration of a single dose of cebranopadol at either 400 mcg or 600 mcg provided superior analgesic effects compared to 60 mg controlled-release morphine, as measured by SPI_2–10_ (sum of pain intensity 2–10 h after administration). While both cebranopadol and morphine produced sufficient pain relief, cebranopadol was better tolerated, and patients reported higher overall satisfaction, with the 400 mcg dose leading to less nausea, vomiting, and dizziness [38].

These potential advantages have also led to cebranopadol being studied for the management of chronic pain. In a phase II multicenter, double-blind RCT, Christoph et al. showed statistically significant improvements from baseline pain for all doses of cebranopadol (200, 400, or 600 mcg once daily) over placebo in 360 patients with moderate to severe chronic lower back pain that completed a 12-week period [39]. Similar results were seen for cancer-related pain as well. In a double-blind RCT of 126 patients, Eerdeckens et al. showed that cebranopadol at 200 to 1000 mcg was effective, safe, and well tolerated over a 7-week period. When looking at the primary endpoint of average daily rescue medication intake, cebranopadol was non-inferior and superior over prolonged-release morphine [84]. Furthermore, safety continues to be demonstrated when following these patients up to 26 weeks [85].

Overall, cebranopadol is a well-tolerated, effective analgesic that demonstrates several potential advantages over conventional opioids, including a reduced side-effect profile and lower abuse potential. According to the drug developer Tris Pharma, it is currently being tested in phase III clinical trials and has been granted fast-track status from the FDA [86].

### 2.4. Dinalbuphine

Dinalbuphine sebacate (DNS), a prodrug of nalbuphine, was released in Taiwan in 2017 as a result of the unmet need for long-acting analgesics. Nalbuphine is a semi-synthetic opioid that has been employed extensively in humans for years. The chemical structure is shown in Figure 4. This mu partial antagonist and potent kappa agonist has been compared to morphine and other opioids for its effective relief of moderate to severe postoperative pain [87]. Available under its brand name Nubain, nalbuphine has an onset of action between 2 to 3 min after IV administration and an onset of less than 15 min when administered subcutaneously or intramuscularly (IM). It has a plasma half-life of 5 h and an analgesic duration of 3 to 6 h [88]. The short half-life of nalbuphine indicated a need for frequent injections in clinical practice to maintain its analgesic impact.

To counter the limitation of its short half-life, a single injection of 150 mg oil-based-formulation DNS allows for extended release, promising moderate to severe postoperative pain relief for 7 days [40]. DNS is an ester derivative of nalbuphine that requires conversion to its active component by endogenous esterases. The ester prodrug is composed of a sesame oil and benzyl benzoate solution [87].

#### 2.4.1. Clinical Studies

A 2012 clinical study performed in part by Lumosa Therapeutics, the co-developers of DNS, examined the pharmacokinetics of the prodrug and bioavailability of the active drug in 12 healthy volunteers. After an IM injection of the DNS formulation, the relative bioavailability compared to nalbuphine HCl was 85.4%, and the mean absorption time of nalbuphine from DNS was 145.2 h. It took approximately 6 days for DNS to fully enter the blood stream where it would be rapidly hydrolyzed into nalbuphine to exert its analgesic effect [87].

A second clinical study sponsored by Lumosa Therapeutics was trialed in 221 patients in 2015. Examining DNS injection against a placebo, this phase III multicenter, double-blind study showed that patients who received a single IM DNS injection of 150 mg 12 to 36 h before hemorrhoidectomy experienced significant reduction in cumulative pain intensity from the end of surgery through to 48 h and 7 days after surgery [40]. A cumulative 62.4% of total patients (77.1% of DNS recipients and 88.3% of placebo recipients) reported at least one adverse event. Most were mild and deemed unrelated to drug treatment. However, there were two incidences of moderate fever in the DNS group. The most common drug-related adverse events reported included pyrexia, dizziness, vomiting, nausea, abdominal pain, headache, and somnolence [40].

In a phase II/III double-blind RCT of 60 patients undergoing laparoscopic bariatric surgery, Lee et al. found that DNS could be a valuable addition to standard analgesic treatment. In addition to a standard multimodal anesthetic approach of parecoxib, propacetamol, and morphine at the end of the operation, an injection of 150 mg DNS into gluteus muscles 12 h prior to the operation yielded significantly reduced pain intensity for 48 h after surgery. Specifically, the number of patients who reported moderate to severe pain postoperatively was reduced by 66% [41]. In addition, a phase IV open-label RCT by Chang et al. in 107 patients undergoing laparotomy also demonstrated that a single dose of 150 mg DNS injected one day before the surgery was more effective than IV fentanyl PCA in reducing pain intensity at 4, 24, 32, 72, 120, and 144 h after the surgery and improving quality of life [42].

#### 2.4.2. Potential Advantages

Multiple studies have compared the safety and efficacy of active drug nalbuphine with that of morphine, supporting the postulation that nalbuphine may present a significant decrease in adverse reactions. A meta-analysis by Zeng et al. comparing nalbuphine and morphine in 15 trials and 820 patients found that while the analgesic efficacy of morphine and nalbuphine are comparable, nalbuphine demonstrated a more impressive record of safety with a lower incidence of pruritus and respiratory depression [89]. However, nalbuphine has a short half-life and requires frequent injection. Due to its extended release and rapid hydrolysis, DNS establishes a lower maximum blood concentration than nalbuphine, thereby presenting a lower risk of adverse events [90].

As a single injection that can be delivered before surgery, DNS can reduce postoperative opioid use and, thus, potentially the incidence of ORAEs. A retrospective study by Chang et al. in patients undergoing laparotomy for gynecologic cancers showed that patients receiving DNS had not only significantly lower NRS scores on days 1 to 5 but also had decreased postoperative consumption of morphine equivalents compared to IV fentanyl PCA. Additionally, DNS led to significantly faster ambulation [90]. Another retrospective study by Zheng et al. examining the efficacy of DNS as part of a multimodal analgesia regimen (parecoxib, acetaminophen, and celecoxib) in upper extremity trauma surgery showed that the proportion of patients requiring opioids for postoperative breakthrough pain was significantly lower in those treated with DNS [91]. Finally, Lee et al. found that there was no statistically significant difference in pain scores when comparing preoperative DNS to postoperative morphine in a randomized study of 43 patients following laparoscopic cholecystectomy [92].

### 2.5. Dual Enkephalinase Inhibitors (STR-324, PL37, PL265)

Met- and leu-enkephalins are endogenous opioid ligands that activate mu- and delta-opioid receptors (with stronger affinity for delta receptors) to produce analgesia and other effects [7]. Dual enkephalinase inhibitors (DENKIs), such as STR-324, PL37, and PL265, work by blocking both of the two major enkephalin-degrading enzymes (neprilysin and aminopeptidase N). The chemical structures of DENKIs are shown in Figure 5. This inhibition results in higher synaptic concentrations of the signaling molecules and increased stimulation of opioid receptors at both central and peripheral receptors [93]. DENKIs have been shown to produce analgesic effects in animal models and preliminary human studies with fewer adverse effects than conventional opioids [94].

There are certain disadvantages associated with exogenous opioids. Drugs administered externally do not target specific locations or achieve the same concentrations as naturally occurring ligands within the body. Additionally, the levels of endogenous neurotransmitters in the synaptic system are closely regulated through homeostasis. As such, external ligands may disturb normal control mechanisms [94]. Theoretically, inhibition of the degradation of endogenous opioid ligands can avoid some of the shortfalls of exogenous opioids and produce a more natural analgesic effect.

#### 2.5.1. Preclinical Studies

Despite the limited number of human studies conducted on DENKIs, preclinical investigations show encouraging outcomes suggesting that DENKIs PL37, PL265, and STR-324 may provide efficient pain relief while minimizing specific side effects typically linked to conventional opioids.

Preclinical research on the DENKI PL37 has shown promising results in various models of pain and migraine. Studies conducted on mice and rats demonstrated that PL37 administration, either IV or orally (PO), effectively attenuated stress-induced periorbital hypersensitivity, facial grimace responses, and cutaneous hypersensitivity [95,96]. Furthermore, PL37 exhibited antinociceptive and anti-allodynic effects in a rat model of peripheral neuropathic pain [96]. It also suppressed osteosarcoma-induced thermal hyperalgesia in mice when administered PO [93]. Menendez et al. concluded that PL37 activated micro-opioid receptors because administration of cyprodime, a specific antagonist of the micro-opioid receptor, inhibited antihyperalgesic effects [93].

Topical instillations of PL265 significantly decreased corneal inflammation in a corneal inflammatory pain model [97]. Additionally, PL265 demonstrated preventive and alleviative effects in a murine model of neuropathic pain, acting specifically at the level of peripheral nociceptors. The repeated administration of PL265 did not induce tolerance, making it a promising approach to prevent and alleviate neuropathic pain without the unwanted effects of traditional opiates [98]. Furthermore, PL265 exhibited long-lasting oral analgesic effects, and its efficacy was mediated through stimulation of peripheral opioid receptors as demonstrated by naloxone methiodide reversion [99].

In a rat model of mononeuropathy, continuous IV administration of STR-324 over seven days significantly reduced pain-related behavior, along with the pain-evoked expression of spinal c-Fos, demonstrating that the drug acts at least in part through inhibition of endogenous nociceptive pathways [100]. STR-324 infusion also resulted in reduced responses to mechanical stimuli, with its antinociceptive effect being reversed by naloxone, an opioid antagonist [101]. Notably, during the three-day postoperative period, no adverse effects on respiratory rate, oxygen saturation, arterial pressure, or heart rate were induced by opiorphin [101].

#### 2.5.2. Clinical Studies

The only published DENKI trial in humans to date is a randomized, double-blind, placebo-controlled ascending dosing study in 78 volunteers to evaluate the safety, tolerability, pharmacokinetics, and pharmacodynamics of STR-324 [43]. Data from the study suggest a favorable safety and tolerability profile in healthy males up to a dose of 11.475 mg per hour administered through a 48 h infusion. While some adverse effects occurred during the course of the experiment, no dose-dependent effect was observed in relation to STR-324. The exact pharmacokinetic parameters of STR-324 were not determined given the difficulty in measuring quantifiable concentrations in the plasma. However, the team did conclude that STR-324 undergoes rapid metabolism and quickly distributes beyond the bloodstream after administration.

However, no consistent antinociceptive action of STR-324 was observed during the study. Moss et al. postulate that this may be due to decreased involvement of endogenous enkephalinases in the acute pain models used. It is possible that the PainCart-evoked pain model does not create sufficient pain intensity for an effect of STR-324 to be shown or that enkephalinases are only significantly active in prolonged pain [43].

DENKIs have shown promising preclinical results with regards to anti-nociception, tolerability, and reduced side-effect profiles compared to conventional opioids. Further clinical research is necessary to determine the effectiveness, dosing, pharmacokinetic parameters, and side-effect profiles of DENKIs in humans. To date, several clinical trials have been registered by pharmaceutical companies including PL37 (phases I and II) and PL265 (phase II) by Pharmaleads as well as STR-324 in phase II by Stragen [102,103,104]. These studies indicate that DENKIs are generally well tolerated and safe in humans through PO administration.

### 2.6. Endomorphin-1 Analog (CYT-1010)

Endomorphin-1 is an endogenous opioid that produces powerful antinociception with fewer side effects than opioid alkaloids in rodent models. Because endomorphins are small peptides, their metabolic instability and poor penetration of the blood–brain barrier and gastrointestinal mucosa make them insufficient for clinical use [105]. While effective in treating neuropathic pain, endomorphins are known to produce adverse effects in the urinary, gastrointestinal, cardiovascular, and respiratory systems [106]. The discovery of endomorphins in 1997 sparked the development of endomorphin analogs to improve permeability and prevent enzymatic degradation as well as to potentially mitigate side effects [105].

CYT-1010 is one such analog, shown to be a highly selective agonist of the MOR [106]. CYT-1010 is a cyclized, D-lysine analog with a novel mechanism of action targeting traditional mu- and exon 11/truncated mu-opioid receptor 6TM variants [24]; its chemical structure is shown in Figure 6. It has inhibitory constants of 0.25 nM at MOR, 38 nM at delta-opioid receptors, and 248 nM at kappa-opioid receptors. While endomorphin-1 has an in vitro plasma half-life of six minutes, CYT-1010 had no observable degradation after four days of incubation. CYT-1010 has shown enduring in vitro stability in laboratory tests conducted on animal serum and plasma as well as in the membranes of both animal and human hepatocytes. These results indicate that the analog possesses the necessary metabolic stability to deliver via IV or oral therapy [24].

#### Clinical Studies and Potential Advantages

The biopharmaceutical development company Cytogel Pharma placed CYT-1010 through phase I testing in 2011. In the initial study, patients tolerated all dosages of CYT-1010, and a dose-limiting toxicity was not found. The adverse effects reported included dizziness, flushing, and transient tachycardia, which increased dose-dependently. The adverse events were all mild to moderate in severity. There were no serum chemistry, hematology, or coagulation abnormalities. According to the company, the FDA has approved Cytogel Pharma’s study protocol for phase II testing [107]. Comparing CYT-1010 to conventional opioids, the phase I study observed no respiratory depression and no decrease in plasma oxygen saturation or change in respiratory rate over the first three hours when compared to placebo [24].

CYT-1010 has also demonstrated reduced abuse potential in preclinical rodent trials. When administered at equipotent doses, rodents treated with morphine spent a significantly increased time in a chamber paired with morphine, whereas CYT-1010-treated mice had no conditioned preference [24]. Furthermore, mice treated with endomorphin analogs exhibited decreased respiratory depression at dosages that produced a longer duration of antinociception than that produced by morphine. Motor coordination was also found to be impaired by morphine but not by endomorphin analogs [108]. Whether this lower abuse potential would be reflected in clinical settings remains to be seen.

## 3. Conclusions

Adequate control of acute postoperative pain is crucial for good patient outcomes in those undergoing surgery. While traditional mu receptor agonists (e.g., morphine, oxycodone) have been most commonly used in managing postoperative pain, they come with the risk of a myriad of undesirable ORAEs such as nausea, vomiting, sedation, respiratory depression, and even abuse potential. The rise of novel opioids with alternative mechanisms of action may serve as potential alternatives due to their ability to provide potent analgesia while minimizing adverse side effects. Further advances in drug–drug interaction prediction tools can help aid future drug development by predicting the effects of drug metabolism and determining dosing recommendations [109].

The more established novel opioids presented include oliceridine, tapentadol, and cebranopadol. Oliceridine, a biased ligand that selectively activates G-protein signaling to promote analgesia while downregulating the β-arrestin pathway associated with ORAEs, has shown analgesic efficacy comparable to morphine as well as a lower incidence of gastrointestinal effects and respiratory depression. Tapentadol with its dual MOR agonist activity and NE reuptake inhibitory effect can synergistically target both nociceptive and neuropathic pain without increased burden of adverse effects. Cebranopadol, a MOR agonist with NOP activity, can produce analgesia at a similar (if not higher) level to morphine with better tolerability for patients.

On the other hand, other novel opioids currently still in development in the U.S. are DNS, DENKI, and endomorphin-1 analogs. DNS, a mu partial antagonist and kappa agonist, has already demonstrated analgesic efficacy in clinical trials conducted in Taiwan as a single intramuscular injection that is easy to administer in patients. DENKIs, such as STR-324, PL37, and PL265, inhibit two major enkephalin-degrading enzymes, thereby enhancing stimulation of both central and peripheral opioid receptors [41]. They have shown strong analgesic effects in preclinical studies and are currently being investigated in clinical trials for efficacy, safety, and tolerability. Finally, CYT-1010 is an endomorphin-1 analog with highly selective MOR agonist activity that has demonstrated good tolerability in phase I trials and even reduced abuse potential in preclinical rodent models. Future studies on its analgesic efficacy and dosing are currently in development. Overall, by providing equipotent analgesia while reducing adverse effects, novel opioids may offer new advantages over traditional opioids to effectively treat and manage acute postoperative pain.

## Figures and Tables

**Figure 1 pharmaceuticals-17-00029-f001:**
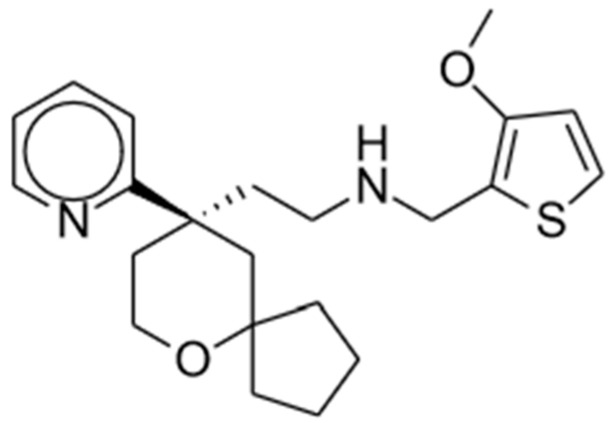
Oliceridine.

**Figure 2 pharmaceuticals-17-00029-f002:**
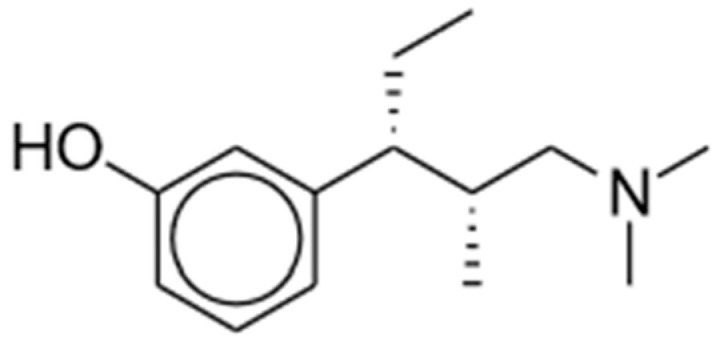
Tapentadol.

**Figure 3 pharmaceuticals-17-00029-f003:**
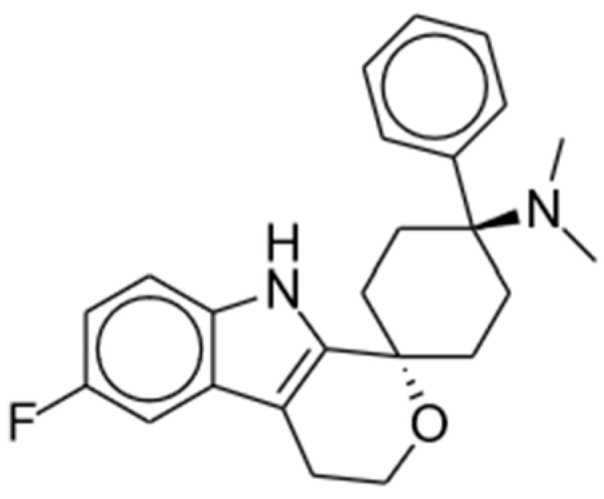
Cebranopadol.

**Figure 4 pharmaceuticals-17-00029-f004:**
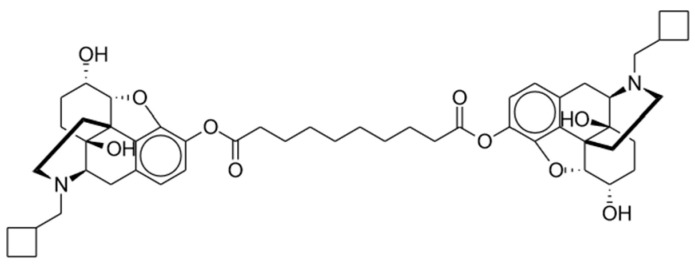
Dinalbuphine sebacate.

**Figure 5 pharmaceuticals-17-00029-f005:**
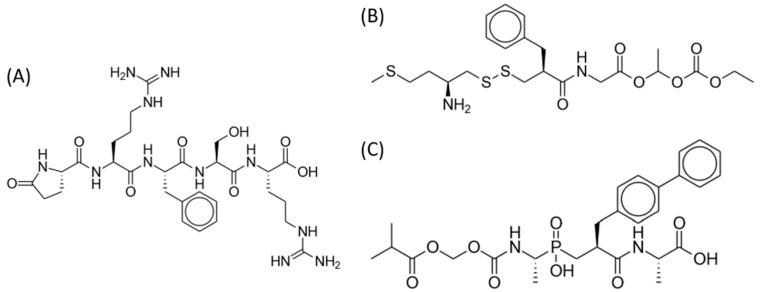
Dual enkephalinase inhibitors. (**A**) STR-324. (**B**) PL37. (**C**) PL265.

**Figure 6 pharmaceuticals-17-00029-f006:**
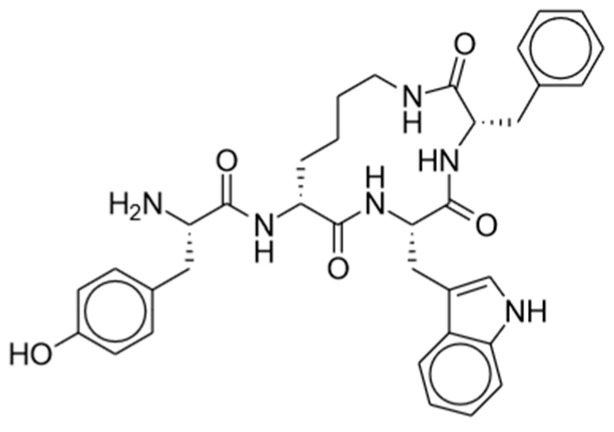
CYT-1010.

**Table 1 pharmaceuticals-17-00029-t001:** Summary of novel opioids.

Novel Compound and Structure	Mechanism of Action	Selected Clinical Trials and Studies
Oliceridine 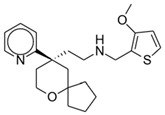	Mu-opioid agonist	Soergel (2014), Phase I [30]: Oliceridine (3 mg) produced greater analgesia compared to morphine with lower incidences of reduced respiratory drive and severe nausea.
		Viscusi (2016), Phase II [31]: Oliceridine (2 and 3 mg) after bunionectomy significantly lowered pain score compared to placebo, with the 3 mg dose having a significant improvement over morphine while showing similar tolerability.
		APOLLO-1 & 2 (2019), Phase III [32,33]: Oliceridine is effective as PCA, providing significant relief from moderate/severe postoperative pain compared to placebo with better safety and tolerability vs. morphine.
		ATHENA (2019), Phase III [34]: Oliceridine is safe and well tolerated in patients (both postoperative surgical and non-surgical) with moderate/severe acute pain.
Tapentadol 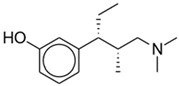	Mu-opioid agonist	Wang (2020), Systematic Review and Meta-Analysis including 8 RCTs [35]: High-dose tapentadol IR (75–100 mg) is as effective as other opioids for acute pain and is associated with fewer GI adverse effects.
	Norepinephrine reuptake inhibitor	
Cebranopadol 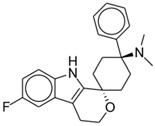	Mu-opioid agonist	Dahan (2017), Phase I [36]: Cebranopadol (600 mcg) produced respiratory depression but not apnea (as seen in full mu-opioid receptor agonists).
	Nociceptin opioid peptide receptor agonist	Göhler (2019), Phase I [37]: Cebranopadol has lower abuse potential than hydromorphone IR.
	Some delta- and kappa-opioid agonist activity	Scholz (2018), Phase IIa [38]: Single-dose cebranopadol (400 and 600 mcg) produced more effective analgesia after bunionectomy compared to morphine with better tolerability and patient satisfaction rating.
		Christoph (2017), Phase II [39]: Cebranopadol demonstrated statistically significant and clinically relevant analgesia for chronic lower back pain compared to placebo with acceptable tolerability.
Dinalbuphine sebacate (DNS) 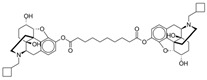	Partial mu-opioid antagonist	Yeh (2017), Phase III [40]: Single-dose DNS (150 mg) before hemorrhoidectomy produced significant reduction in cumulative pain intensity postoperatively.
	Kappa-opioid agonist	Lee (2023), Phase II/III [41]: In addition to standard perioperative multimodal analgesia, single-dose DNS (150 mg) allows significantly better pain relief 48 h after laparoscopic bariatric surgery.
		Chang (2020), Phase IV [42]: Single-dose DNS (150 mg) resulted in lower pain intensity compared to IV PCA with fentanyl.
STR 324 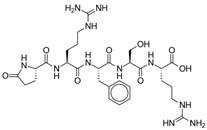	Dual enkephalinase inhibitor	Moss (2022), Phase I [43]: STR-324 showed favorable safety and tolerability profiles at doses up to 11.475 mg/h in healthy male subjects.
CYT-1010 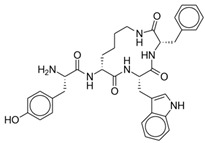	Endomorphin-1 analog with selective mu-opioid agonism	Webster (2020), Phase I [24]: CYT-1010 demonstrated significant analgesia and no respiratory depression or decrease in plasma oxygen saturation at dose levels tested.

## Data Availability

No new data were created or analyzed in this study. Data sharing is not applicable to this article.

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
