# Peer review of "Novel Opioids in the Setting of Acute Postoperative Pain: A Narrative Review"

_pharmaceuticals, 2023, doi:10.3390/ph17010029_

Round 1

Reviewer 1 Report

Comments and Suggestions for Authors

1. More clinical data are needed to clarify the impact of differences in drug metabolism on drug efficacy. Will drug concentrations that are too high or too low necessarily affect drug efficacy?

2. I hope to supplement it with reference to the following clinical data in an article. New opportunities and challenges in natural product research: When target identification meets single-cell multi-omics, this article summarizes the research progress on the efficacy and mechanism of important natural products in inflammatory diseases and malignant tumors, and especially proposes a large variety of traditional Chinese medicines. Efficacy evidence evaluation system. 2022 proprietary Chinese medicines related to 40 inflammatory diseases and cancers were evaluated from 105 dimensions, including non-clinical animal experiments, clinical randomized controlled trials, retrospective or real-world studies, human experience, treatment guidelines or diagnosis and treatment plans.

References are as follows: New opportunities and challenges in natural product research: When target identification encounters single-cell multi-omics Zhu Yuyu, Ouyang Zijun, Du Haojie, Wang Meijing, Wang Jiaojiao, Sun Haiyan, Kong Lingdong, Xu Qiang*, Ma Hongyue *, Sun Yang*Journal of Pharmaceutical Sciences B 10. https://doi.org/1016.2022/j.apsb.08.022.<>

3. Although drug metabolism gene testing technology can help doctors make more reasonable clinical decisions, its high cost is unacceptable. It should be noted that the DDI-Predictor tool can replace the accuracy of predicting drug metabolism.

Comments on the Quality of English Language

Minor editing required for English

Author Response

Dear Reviewer, 

Thank you for your insightful comments on the need for more clinical data to evaluate dose response. The concentrations of the novel opioids used in clinical studies are mentioned, including any dose-dependent effects on efficacy if evaluated.

We appreciate your recommended references on traditional Chinese medicines and related natural products. Added how naturally derived products show promise in the development of novel compounds.

We also noted how DDI predictor tools can aid in future drug development by predicting drug metabolism and recommending doses.

We look forward to hearing from you. 

Yours Sincerely

Reviewer 2 Report

Comments and Suggestions for Authors

Postoperative pain control is very important in all areas of surgery. Good pain control will result in minimal postoperative complications and high patient satisfaction. This article, which evaluates the properties and mechanisms of action of drugs used in postoperative pain control, may be published as a version of this article. 

Kind regards.

Author Response

Dear Reviewer:

We appreciate your kind words and your expertise. 

Yours Sincerely

Reviewer 3 Report

Comments and Suggestions for Authors

1. The manuscript seems to be monotonous.

2. There are no structures of novel opioid drawn throughout the manuscript.

3. The manuscript would be more informative if the text are represented as Figures and Tables.

4. The mechanism of actions of novel opioid must be represented by Figure of Diagram

5. I think the manuscript is not suitable for publication in "Pharmaceuticals"

6. All the compounds are of known structure and their structure must be included in the manuscript.

7. An overview of various signaling pathways should be represented in diagrams to represent the mechanism of action.

8. Ki and EC50 should also be added for the novel opioid analogues against their molecular target (μ-opioid receptor, δ-opioid receptor, and κ-opioid receptor).

9. Authors should also explore the other novel opioids in development stages DNS, 625 DENKI, and endomorphin-1 analogs should be added in the manuscript.

Recommendation: Reject and encourage resubmission

Author Response

Dear Reviewer,

Thank you for your comments, please see a point-by-point response posted below. 

1. The manuscript seems to be monotonous.

We have attempted to address the language and flow of the manuscript.

2. There are no structures of novel opioid drawn throughout the manuscript.

We have now included the chemical structures of the novel opioids presented throughout the manuscript.

3. The manuscript would be more informative if the text are represented as Figures and Tables.

We have summarized the novel opioids, their mechanism of action and key studies into a table. 

4. The mechanism of actions of novel opioid must be represented by Figure of Diagram

This is now covered in table 1

5. I think the manuscript is not suitable for publication in "Pharmaceuticals"

We acknowledge your comments

6. All the compounds are of known structure and their structure must be included in the manuscript.

This is included in table 1

7. An overview of various signaling pathways should be represented in diagrams to represent the mechanism of action.

We have now included a table summarizing the novel opioids presented, including their chemical structures, mechanisms of actions, and selected clinical trials. However, this narrative review focuses on the clinical trials to date rather than detailed mechanism of action.

8. Ki and EC50 should also be added for the novel opioid analogues against their molecular target (μ-opioid receptor, δ-opioid receptor, and κ-opioid receptor).

We have added Ki and EC50 data added where available.

9. Authors should also explore the other novel opioids in development stages DNS, 625 DENKI, and endomorphin-1 analogs should be added in the manuscript.

DNS, DENKIs, and endomorphin-1 analogs discussed in sections 2.4, 2.5, and 2.6.

Thank you for your expertise and we look forward to hearing from you.

Reviewer 4 Report

Comments and Suggestions for Authors

General comments:

The paper presents a review study on novel opioids in the setting of acute postoperative pain.

The research is relevant both for research and practice.

The manuscript is well written and understandable.

The paper could nicely fit into the Special Issue New Endogenous Opioid Peptides and Peptidomimetics with Potential Biological Importance.

Specific comments:

The authors may clarify in more detail how the PRISMA guidelines where followed.

Differences regarding the quality across included studies may be more emphasized.

The authors may want to check if more detailed analyses using meta-analytical approaches could be feasible. It would strengthen the discussion of findings.

Important variables need to be discussed in more depth, including the role of sex differences, age differences, source of pain, region of pain, etc.

The cost-effectiveness ratio could be better specified.

The practical relevance may be illustrated with a broader set of examples.

Author Response

Dear Reviewer,

Thank you for your expertise and comments, please see the point-by-point response below. 

The paper presents a review study on novel opioids in the setting of acute postoperative pain.

The research is relevant both for research and practice.

The manuscript is well written and understandable.

The paper could nicely fit into the Special Issue New Endogenous Opioid Peptides and Peptidomimetics with Potential Biological Importance.

Thank you for your kind words

Specific comments:

The authors may clarify in more detail how the PRISMA guidelines were followed.

This manuscript is intended as a narrative review, rather than a meta-analysis/systematic review. The title has been adjusted to reflect this. 

Differences regarding the quality across included studies may be more emphasized.

Thank you, we have added more comments on key study design features (number of patients, blinding, multicenter, etc) of the more important studies for each drug.

The authors may want to check if more detailed analyses using meta-analytical approaches could be feasible. It would strengthen the discussion of findings.

Thank you for your feedback. Unfortunately the heterogeneity in clinical setting, administration regimen, and outcome measurements makes it challenging to conduct a high-quality meta-analysis. Where possible, we have commented on comparisons between different studies. 

Important variables need to be discussed in more depth, including the role of sex differences, age differences, source of pain, region of pain, etc.

Different variables discussed at the end of section 1.1.5. Added discussion of sex and age differences and the need for more studies.

The cost-effectiveness ratio could be better specified.

We have Added discussion on the cost-effectiveness of novel opioids with examples of the incidence and cost of opioid-related adverse effects.

The practical relevance may be illustrated with a broader set of examples.

 In addition, to examples of potential advantages mentioned in each novel opioid section, added broader example of how a targeted approach of using novel opioids in those with risk factors for opioid-related adverse effects, such as advanced age, obesity, and sleep apnea, may be more practical given that morphine is cheaper.

Thank you and we look forward to hearing from you. 

Yours sincerely

Round 2

Reviewer 3 Report

Comments and Suggestions for Authors

Manuscript is ok now